# A Bird with No Name Was Born, Then Gone: A Child’s Processing of Early Adoption through Art Therapy

**DOI:** 10.3390/children10040751

**Published:** 2023-04-20

**Authors:** Einat S. Metzl

**Affiliations:** Department of Jewish Art (Art Therapy Graduate Program), Bar-Ilan University, Ramat-Gan 5290002, Israel; einat.metzl@biu.ac.il or emetzl@yahoo.com; Tel.: +972-546-879-676

**Keywords:** adoption, art therapy, symbolic processing, attachment, expressive

## Abstract

An art-based case study was used to illustrate the therapy journey of a child working through issues related to early adoption. The objective of this case was to review art products and clinical notes systematically, exploring main clinical themes and illustrating both challenges related to adoption and the potential of art therapy to support healing within this context. The methods of investigation and report focused on exploring the meaning of narratives, the art products, and the relational dynamic which emerged during sessions. The results are discussed within the context of the relevant literature, emphasizing considerations for working through adoption challenges in art therapy.

## 1. Introduction

As an art therapist who is also trained as a family and couples therapist, I am moved by witnessing and shepherding many voyages. At times, some stories beautifully intertwine with our own, profoundly anchoring us to our work. My work with Elizabeth (pseudonym) and her family was such an experience. It recounts a therapy course she undertook several years ago from age 9 to age 11. The art included was carefully chosen (with permission from the family) to illuminate themes that children in my practice dealing with parent loss, adoption, or attachment often work through. The dynamic narrative and artwork are presented after a brief review of the literature related to adoption, attachment-based art therapy, and the art therapy conceptualizations for children who are/were adopted [1]. Following the case illustration, a discussion and conclusion seek to underscore the main clinical considerations related to art therapy interventions with children who are navigating adoption journeys.

### 1.1. Adoption as a Traumatic Event

Currently, seven million US residents are adopted and 1.5 million of those are adopted children [2]. About 250,000 children in the United States alone are in the foster care system and will either be placed with a family or age out of foster care when they turn 18 [3]. Being adopted constitutes a uniquely challenging experience, one that shakes the core assertion of who you are, who and where you belong, and how desired and worthy you are at birth [1,4]. Nevertheless, circumstances around an adoption can greatly impact how traumatic a given situation is, often directing therapists to focus on either a trauma-focused approach or an attachment-based approach.

Malloy [5], for example, surveys the crucial clinical and ethical considerations such circumstances present. For example, she examined how the permanency placing of children in foster care in the United States might have compounded their experience of ongoing instability, undesirability, lack of secure attachment, and having a limited voice in the system guiding their immediate family situation. Issues related to secrecy, shame, and limited communication around the process often exacerbate the traumatic, pre-verbal experience of children trying to navigate who they are [4,6]. As Ryan [7] concluded after researching the experiences of post-adoption families, “Mental health concerns among adopted children and adolescents are common and complex, and that parental stress and need for post-adoption support is high” (p. 215).

Children impacted by major traumas often present with trauma responses of hypo- or hyper-responding to their surroundings, signs, and symptoms that Levine and Kline [8] have termed “the universal symptoms of trauma” (p. 40). These indicators may present differently at different ages, and there may be secondary trauma symptoms that are specific to different traumas [8]. Being adopted, for example, can provoke secondary trauma of socio-emotional adjustment to new environments, challenges with attachment and trust, marginalization (perceived and/or actual) within their home and school environments, and so forth [9,10]. During the attachment-forming years, one’s distress may also play out behaviorally with caretakers, impacting already fragile attachment formation processes [11,12].

### 1.2. Adoption as a Psycho-Socio-Developmental Challenge

In working with children who have been adopted, one must understand the developmental and psychological merit of attachment, and the impact of trauma and loss on a developing child [5,13]. The child’s sense of self and belonging forms within the interplay of the re-formed family, his/her developing mind and body, and a plethora of socio-environmental narratives about the meaning and value of the adoption. So, in addition to the attachment and developmental tasks of every childhood, the adopted child must contend with an attachment trauma that can also trigger systemic disruptions [14]. For example, how does a neighbor, a grandparent, or a friend view the adoption? How do they interpret the child’s responses to his/her adopting parents? Are the motives of the parents giving up the child for adoption or of the adopting parents questioned? All these factors inform the family’s narrative and can support or inhibit the child’s psycho-socio-development. In addition, the influence of the adopting family unit cannot be understated; it is where repair happens, and where distrust and attachment injuries (and often attachment reactivity, ambivalence, and avoidance) try the adoptive parents and siblings [15]. In this case, working through steps of building trust moves through disruption and repair first with the therapist, and then illustrates the way art interventions “function as a facilitator for family integration” [6] (p. 29), first in the daughter–mother dyad and later in the father–daughter dyad.

### 1.3. Art Therapy with Children Who Have Been Adopted

Art therapy is a mental health profession that enriches the lives of individuals, families, and communities through active art making, the creative process, applied psychological theory and human experience within a psychotherapeutic relationship” [16], as proposed in “*About Art Therapy*”. Over the years, art therapy interventions have been developed to support the specific needs of different populations, following the above principles and particularizing the use of art through different materials and interventions [17]. Art therapy interventions offer a unique therapeutic holding space for children in foster care or who have been adopted; for examples, see [4,6,18]. In art therapy, psychological material can be at once concrete and symbolic, have both a “here-and-now” impact and lasting reminder of the felt experience [19] and an internal sense of mess and shame [20]. Gil and Rubin [21] have highlighted the importance of informing and enhancing therapists’ self-awareness through play and art making with children in general, and with children for whom attachment challenges are part of the original trauma in particular. When working with children who have lost a parent/parents to any circumstances, and whether new parents have assumed the parenting role, the transferential relationship with the therapist is essential to processing the original loss and attachment insecurity. After a “good-enough” attachment is worked through with the therapist, who has gained an internal sense of the child’s inner life, the therapist can utilize art-making and play to support parent–child dyads [12]. In those spaces, the art therapist becomes an adult ally of the family [1,19], helping to express and process unconscious and pre-verbal experiences, transforming them from barriers into connecting opportunities [22]. Each child has a unique experience with adoption, depending upon his/her cultural/racial affiliation, class, family, and socio-educational communities [23], spaces in which sectioned expressions are expected and desired. Art therapy can be a liberating space for children who are unsure of their place, as it offers playful and expressive opportunities to experiment with attachment and belonging outside of complicated familial settings, and with a caring adult, responding to their needs in a set, secure space [19]. Several art therapists [4,6,14,15,18,20,24,25,26] similarly suggested that children and teens who can work through attachment disruptions, identity questions, and emotional challenges related to their attachment issues through art therapy. Robertson [4], for example developed an eight-themed journal intervention in which teens explore (1) their original script, (2) missing pieces, (3) loss and impermanence, (4) self-portrait of inherited traits, (5) being singled out for being adopted, (6) adoptive family portrait, (7) the “Ally” within, and (8) the bridge to new connections. The case presented here offers a similar arc, but with a younger child, and focuses on a more relational/family therapy approach.

### 1.4. Something to Hold on to: The Importance of Metaphors and Transitional Objects

Case [24,25] explored the use of art materials, art making, and art play as uniquely profound tools for children to explore, display, and manage chaos; examine their authentic experiences and merit; and build and destroy aspects of themselves. Play and art-based metaphors operate as safe representations of the self with which the therapist and parents can interact [7] as well as transitional objects imbued with moments of achievement, attunement, and connection [5]. In addition, different art materials and interventions offer engagements at different levels (tactile, sensorimotor, emotional, cognitive, etc.), helping the child, as well as the child–parent dyad, move toward more integrative experiences of self and others (e.g., refs. [15,26]). The use of materials and choice of materials are thus part of the metaphorical communication regarding the child’s internal world [17,19,27]. Attuned attention to the art materials used, the art-making processes, and art products thus support the therapist’s assessment of both children’s and parents’ strengths and challenges [15,17,19,27,28].

## 2. Materials and Methods

### 2.1. Case Study as a Methodology 

The case study as a methodology offers a systematic form of investigation that helps therapists examine and develop clinical practice and art-based interventions [17,29] through observing, acting, reflecting, refining, and developing a deepened understanding of the clinical processes and where and how they are useful in practice. In art therapy, the centrality of art making and reflecting on the meaning of art during and after the sessions also offers a profound anchor for deep integrative learning post-process through the emergent themes [30]. In this case, I reread the clinical notes of all relevant clinical cases, explored the art responses, and identified five clinical themes, which I categorized according to the following titles (by order of occurrence): into the pain, from deep inside, art as nurturance, emerging, and separating. The next section illustrates these clinical themes through Elizabeth’s art voyage in art therapy.

The case study often uses first-person exposure of both client and therapist as sources of knowledge and meaning [29,30] while contextualizing the material that is generalizable to other clients and situations within the particular and psycho-socio-political realities of the case [31,32]. The use of art as a symbolic and concrete externalization of meaning is similarly both personal and relatable, and offered here to deepen readers’ understanding of the process.

### 2.2. Case Background

In Elizabeth’s story, for example, all of the above challenges were palpable, yet her reality was also a good one; she was adopted at birth by loving parents who brought her to therapy and offered her a stable and healthy environment in which to develop. For Elizabeth, being adopted was a fact of life, communicated openly within and outside the home. There was visibility to her adoption as she was of a different racial–cultural background than that of her adoptive parents, circumstances I assumed were both challenging (no hiding that she was adopted) and protective (congruent with her internal sense of self as different). She and her family had ample resources (emotional, mental, relational, and financial) to create as healthy a narrative as possible around how and why she was adopted. She nevertheless seemed to struggle with questions about her identity and place in the family, which played out in frequent anger outbursts that became destructive or even violent at times.

## 3. Case Illustration (Results)

Elizabeth’s mother reached out seeking therapy for her 8-year-old daughter due to anger and anxiety issues, which presented through tantrums, hypersensitivity, isolation, anxiety as well as curiosity and exploration of her adoption story. Elizabeth’s mother hoped that art therapy would provide a space to ease the emotional distress Elizabeth was experiencing, and astutely recognized the timing as likely also connected to the beginning of her daughter’s identity formation and adoption-related issues of grief/loss/trauma/anger. She recognized that Elizabeth did not want to communicate or process very much verbally, yet assumed that the frequent outbursts, emotional liability, and general reactivity had to do with Elizabeth’s renewed questioning about her family of origin. Elizabeth had been adopted at birth, and her adoptive parents often communicated how much they wanted her.

The main treatment goal was to provide a space for Elizabeth to explore, express, and process the feelings and concerns she was having. We also hoped that articulating her needs and wants would replace some of the frustration, hostility, and aggression often directed toward her parents with communication that allowed more validation and connection. In setting up the treatment, I explained my interest in and previous experiences (personally and professionally) with adoptions as well as my method of intervention. Specifically, I work closely with both parents and the child using an attachment-based perspective and art therapy as the main modality. Especially at the beginning, I work individually with the child and meet with the parents monthly to formulate treatment goals and transparently discuss the child’s strengths, expressed needs, and sensitivities, and how those manifest verbally and non-verbally inside the session and outside the session. I later follow the child’s lead as to whether the sessions are individual, parent–child dyads, or full-family sessions if these are appropriate clinically, and support relational repair and systemic integration.

### 3.1. Into the Pain: Forming Attachments despite Failures

When Elizabeth first entered the therapy room, she was visibly uncomfortable; she responded with minimal verbal communication, bold but brief eye contact, and thoroughly surveying the art material and spaces I had to offer. I introduced myself and the intention of our time together. I could see she did not want to speak but was excited to create. She chose materials from the shelves, sat down, and immediately reached for the clay. She created an egg-size oval and held it in her hands, wetting it constantly to smooth it over. It was heavy, she said, and messy, and it needed a place to be. Together, we looked at options for containing and holding it, and she decided we should use my tissue box. So, we took out all the tissues (as she wondered if many children need tissues in my office), and the box housed the clay egg. Then, the clay egg was given feathers and a beak, and a little container of food. A baby bird had appeared inside the tissue box and was hung from the branches of a large plant in my office.

It was the end of the session, and we were both quiet, and engaged and moved by what had emerged. Elizabeth stated she did not want to talk or name the bird. To create closure while symbolizing the importance of the project, we decided to take photos (Figure 1 and Figure 2). We then parted ways until the following week.

As Gil and Rubin [21] suggest, transferential and counter-transferential experiences are communicated directly and profoundly through art and play. In most forms of clinical therapy, the therapist’s and client’s inner worlds correspond and propel felt shifts. In verbal therapy, shifts often include making intuitive and pre-conscious experiences that are realized through intentional verbal communication and physically sharing space. In art and play therapy with children, when attunement occurs between the therapist and client, expressive and active engagement goes around and beneath the words, from one heart to another. An attachment is formed with the therapist as well as with the art process, art materials, and resulting art product.

Bearing witness to Elizabeth’s engagement with the art materials, knowing she was adopted at birth, intertwined with my own—different, yes, but psychologically similar—background story. My mother had passed away when I was 10 months old, and after my dad remarried, we went through a process of adoption by my stepmom, which was dropped midway without an explanation. I share these aspects of myself because the early loss of one’s biological parents—questions around being wanted and not wanted, of deserving parental parent permanence and nurturance—was alive within both Elizabeth and me as she took a chance and placed her heavy clay egg in a box, and then it transformed into a helpless bird. I could sense we both wanted this to symbolize the beginning of a healing journey.

Then, however, something that every art therapist dreads, and had never happened to me before or since, happened. On the day of her second session, I came back to the office, and Elizabeth’s clay bird was gone. The tissue box was thrown on the floor. It must have fallen during the night and someone (Another therapist who shared my office? A person cleaning it?) threw away the clay pieces, which may have rolled out . . . I panicked. I was already deeply connected to the thought that could give a new birth and narrative to this little bird. For me, it was not only a piece of clay or a precious artwork that I was guarding and had lost; the loss threatened to shutter the possibility of forming trusting relationships and taking care of one’s most fragile self-being. I looked everywhere—drawers and trashcans. I called the cleaning service to check if they knew anything, then consulted with a trusted play therapist who knew my work.

Finally, I had to center myself, to face the reality: here she was in my waiting room, and I felt I had failed her. Here, she would re-lose a very young representation of herself. She would re-experience a caring adult’s failure to nurture her. It was now my role to take responsibility. Although I did not intend for it to happen that way, I had to face her with empathy, with acceptance of any response she would have, to reassure her that I am there for her, and to wholeheartedly work toward repair. I quickly printed some photos of the creations we took the previous week and opened the door.

Elizabeth was clearly upset and confused by the disappearance of the little bird. We searched every inch of the office together. I shared with her how sorry I was and how disappointed I was too. I suggested that we use the tissue box that was left, in case she wanted, to create something and write a story of what happened. She agreed, but asked that I write out the following note:

A bird with no name

Was born

Then gone

We placed the note inside the empty tissue box. She placed it on a high shelf (instead of hanging it on my plant) and added a bid, bold sign that said: “DO NOT TOUCH!”

I witnessed the resilient ability of this child to survive and strive; despite the loss, which was deeply felt, she was able to try again, try differently, with a bit more protection. Elizabeth’s box stayed uninhabited for a long while, but we would both look at it between sessions. Then, one day, it became a tree house with a long ladder made from pipe cleaners and was hung back on the plant. However, this was much later in our work together.

### 3.2. From Deep Inside: Making Messes, Tolerating and Exploring Their Meaning

After our first two sessions together, Elizabeth focused for a long time on what seemed to be a very tactile and messy project, concentrating on the here-and-now play process, with little planning or care about the product. She was fully engaged in pouring paint, smearing it, and mixing it with her hands. (See Figure 1, above) At the same time, Elizabeth was clearly checking that I was paying attention, and tested boundaries—would I let her drop paint on my carpet and sofa? Would I allow her to use all of my blue paint? Would I be there when she needed more paint or help cleaning up? Working together then was similar in quality and themes to working with typically much younger children. Similarly to children ages 2 to 3, she was re-growing herself through process art, which is all about sense–perception integration, the exploration of boundaries, and self-regulation in the presence of a good-enough attachment figure. My role was clearly to keep a secure and free-enough space for her to be and to search, emotionally—staying attuned and engaged, appreciative, loving the messy parts, not knowing what might emerge, but trusting what is inside—together (see Figure 3 and Figure 4).

With time, more conversation about intentionality and meaning coincided with the mixing of paint. Elizabeth began to explore the colors she was creating and often matched them with her skin tone. As a darker-skinned person adopted by lighter-skinned parents, she seemed to focus on the meaning of exploring different browns. She would often mix all the colors to create brown, then ask me whether it looked like dirt or feces, was it too messy, and could I see it had all the colors inside it still? As a lighter-skinned therapist, I sensed the transferential relationship forming and her cognitive reflections about her identity being expressed through the art—who was she? How different or similar to her caretakers? Was she messy? Was she inherently dirty? Could she also be clean? Could I/we understand and love her, as she is?

### 3.3. Art as Nurturing

Then, one day, as kids do brilliantly when they have psychologically figured out something, she moved from making messes to making symbols. Predominantly, she focused on letting me know that the art was nurturing for her and asking me to engage with her through art play. So, for example, she created art soup, (clay and paint) meatballs (Figure 1), and an art burrito. She would use some of the tactile play with unstructured materials (paint, wet clay) as before, but completely differently; she maneuvered it with mastery, was proud of the product, and invited me to engage with it (see Figure 5) as I ate or fed her symbolically.

### 3.4. Emerging: Supporting Connection and Validation with Parents

Elizabeth’s parents were always caring and involved. Throughout our individual work sessions, she would choose whether she wanted to have the parent who was picking her up join for the last few minutes to show what she made and help with the transition. Additionally, she knew I had periodic meetings with them to see how things were at home and school, and so forth. As an art therapist and a family therapist, I saw that the most important attachment work was between Elizabeth and her parents, and that art could help support that.

Art was a natural way for Elizabeth to communicate emotional needs, much more so than verbal expression. Art was not the natural form of communication for her parents, however, and so inviting them to join us in the session, where I could help translate some of what I see in the art, seeing the parents’ needs and hers, offered an opportunity for a different connection and attunement. For example, relatively early in the treatment process, when Elizabeth was still really into making messes, she decided to invite her mom to join. As mom began to tell me about struggles over the week, which culminated in a fierce anger outburst that would not subside, Elizabeth began to quickly make circular lines with markers, and then paint a bold black line in the middle, with a smaller white layer on top of it, then add feathers and drips of metallic paint all around. She was clearly expressing her painful recollection of what had happened, sharing with us the many layers of how things evolved for her, between the anger barely contained by the paper and the core struggle of black and white, internal and external. The feathers also reminded me of the young bird we lost after our first session together—another reminder of the pain when parental figures disappoint or fall short, with the metallic tears all around (see Figure 6). As we three looked at the creation together, mom could see and connect the felt sense to the words Elizabeth did not have, but she did. The emotional attunement allowed them to choose a surface on which to place the image, containing it and acknowledging its importance, creating closure through repair.

A similar experience happened later in the treatment when dad joined us. Elizabeth asked him to help “pouring paints when I make a mess.” This was clearly not a comfortable request for him; he wanted to create something he could understand and bring more of his thought and creativity to. To him, the process mostly felt out of control, messy, and potentially costly (if she accidentally spilled paint on my furniture). We discussed his concerns and her wants. I took responsibility as far as the use of materials and time and underscored the meaning for Elizabeth to have him join a discovery process: she is creating for herself on her own terms. I also stressed his role as dad, ultimately allowing for any of this. With that, dad was willing to try. Elizabeth engaged fully in the work, asking him to pour different color paints every time. Initially, she played with the paint with her hands, as she had done with me, but then she tested the setting we created—would we allow her to use her legs to mix the paints, and then would we let her paint them? (See Figure 7) With each step, we considered her wants and risks, and, with my support, dad allowed for the experience. We took pictures to remember the moment, and then we all cleaned up together.

In both parental dyad sessions, a palpable process transpired. Elizabeth tried making messes first with me alone. When she saw that I supported her, she invited her mom to come closer to that part of herself with my negotiation. Correspondingly, after she tested my ability to keep her safe, but allow for free expression and less controlled materials, and give her the time and space she needed in the sessions, she invited dad to learn how to also support those needs.

### 3.5. Individuating/Separating: Creative Experiments with Separating While Staying Connected

The last stage of treatment with Elizabeth illustrates growth, individuation, and expansion of self. She was now creating elaborate mixed-media sculptures of different homes, often made from balsa wood, wires, and fabrics carefully connected with a glue gun. The art materials she used required mastery and problem solving, and her efforts were much more structured and cognitively focused than the initial use of paint and clay [23]. The psychological themes continued to revolve around the finding of new homes. She re-hung the tissue box and created a tree house for a child (no longer a helpless baby bird) that could be accessed by a very long ladder. She then created multiple other homes, discussing different rooms, ways to access them, their safety, and considerations needed for different creatures. Then, she invited dad to join a session; this time she did not need to dictate his role (pouring paint and observing) but was open to co-creating a wooden sculpture (see Figure 8).

Creating together could only happen once Elizabeth had articulated for herself who she was and what she was able to do, and could trust that in the therapy room, her father was there to support her and understand her. From such an individuated stance, the focus on attachment work became an invitation for both to bring themselves more fully, and they were created with joy and competency.

It was time for treatment to end; the intensity and severity of Elizabeth’s anger outbursts had subsided, and she seemed to be doing well at home and school. After a couple of dyadic closing sessions (one with mom and one with dad), and a family session (with the parents and younger sister), she and I had a final termination session. We reviewed her artwork—beginning to finish—and she decided what she wanted to take, leave with me, or discard (after I took photos for clinical records). I then suggested she create something to take with her from my room. The intention was to create a transitional object to carry our connection and her achievements into her home life post-treatment. She immediately asked if she could take a plastic case I had and filled it with a sample of the art materials she had used: markers and feathers, fabrics and a piece of clay, wood and wire. She had many art materials at home, so this was clearly a symbolic gesture, a culminating transition of the “art as nurturance” experience we had. She searched through my large collage materials and found images of three grown birds, and one looking over a nest with eggs (Figure 9). She seemed to represent herself now as a grown bird able to take flight, witnessing her own egg-shaped self, and perhaps also as the egg, with three caring figures (mom, dad, and myself?) surrounding her.

## 4. Discussion

Throughout the course of my work with children, teens, and adults who had been placed in foster care or were adopted, several themes have emerged in the arc of typical treatment, which are often reflected in the literature: 1. delving into the pain, expressing from deep inside, art as a space of nurturance, emerging integration, and separating (from the therapist and therapy). In the case study above, I attempt to illustrate this evolution of themes in art therapy treatment through art examples from Elizabeth’s journey. The effect of art therapy in this case resembles the documented accounts of art therapy for young, traumatized children and adolescents in the literature reviewed [4,6] and underscores the potential noted in the richness of art materials [17,27] and methods of interacting with them [11,13]. For many children who had experienced early attachment trauma, the possibility of “making messes” [11,20,24], experiencing and expressing moments of attunement and raptures of trust [4,8,9], and then surviving those with caring adults—in this case, play and art therapists [19,21,24,25] as well as family members [6,21,27]—support the reintegration of the fractured self into a valuable and authentic self.

Main clinical considerations related to art therapy interventions with children who are navigating adoption, based on the literature reviewed here and the clinical journeys I escorted over the years, include several crucial anchors. First, it seems important for the therapists to become deeply aware of their own attachment sensitivities, family (caretaking) dynamic, and psychological wishes that are likely to play out in the transferential relationship. Second, before working with a child who has been adopted, the original and adoptive family composition, culture, and socio-political narratives as they meet the child’s trauma need to be taken into account [31] when evaluating the fit of the therapist and use of art and therapy for the particular client [32]. Third, and where I attempted to focus this particular case illustration, offering a therapeutic and creative space as a place for the child and her/his family to express, process, and move toward the integration of less and more conscious parts, can offer a profound healing opportunity. In order for the space to offer such an experience, a crucial and gentle balance between structure and freedom, and engagement with traumas and challenges while holding the client with care, needs to be maintained.

This case study has obvious limitations. It tells a particular process of a particular child and family that cannot be directly applied or generalized. It is also limited by my own knowledge, culture, and skills as a therapist and investigator. Nevertheless, it is my hope that this case illustration and the themes that emerged from my clinical experience can further be developed by clinicians working with children who have been adopted or will be more broadly explored through larger studies.

## 5. Conclusions

This paper utilizes a case study to illustrate the ways in which art therapy offers a uniquely powerful tool for clients coping with early attachment traumas. Specifically: (1) art is essential to processing non-verbal, traumatic, or sensory–motor felt senses; (2) art making is a way of speaking directly on multiple levels, connecting differently, and offering new possibilities; (3) art and play are the primary way in which we ALL intuitively experience what we feel and express what we know; (4) art can help us identify feelings and thoughts, and communicate, contain, and express them.

## Figures and Tables

**Figure 1 children-10-00751-f001:**
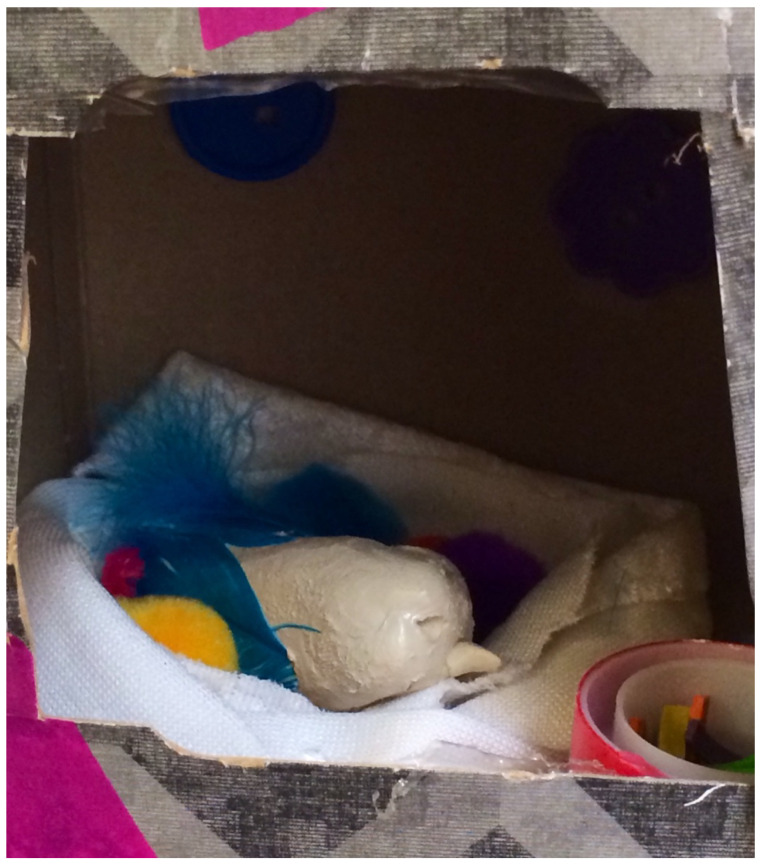
A bird with no name was born. Mixed media (tissue box, clay, feathers, fabric, tape, cut paper, buttons, pompoms).

**Figure 2 children-10-00751-f002:**
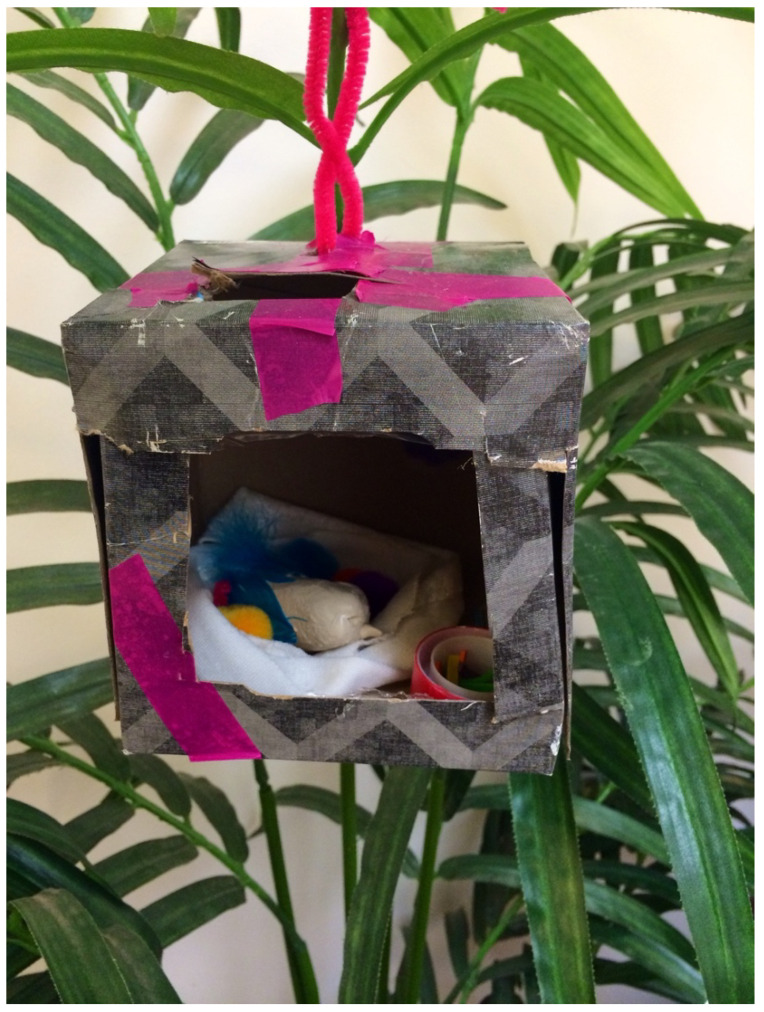
A bird with no name . . . hanging in the therapy room. Mixed media (tissue box, clay, feathers, fabric, tape, paper cut, buttons, pompoms).

**Figure 3 children-10-00751-f003:**
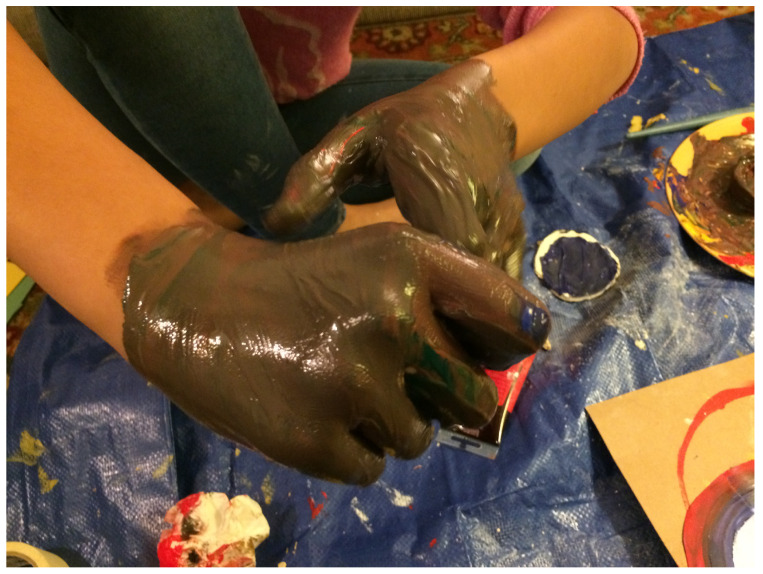
Making and experiencing the mess/Can I be loved. Tempera paints.

**Figure 4 children-10-00751-f004:**
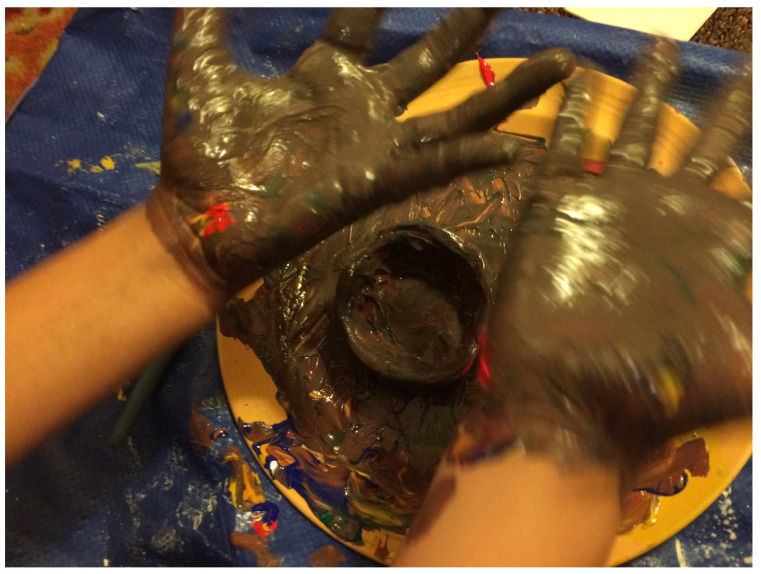
Observing and reflecting/The mess as identity. Tempera paints, clay.

**Figure 5 children-10-00751-f005:**
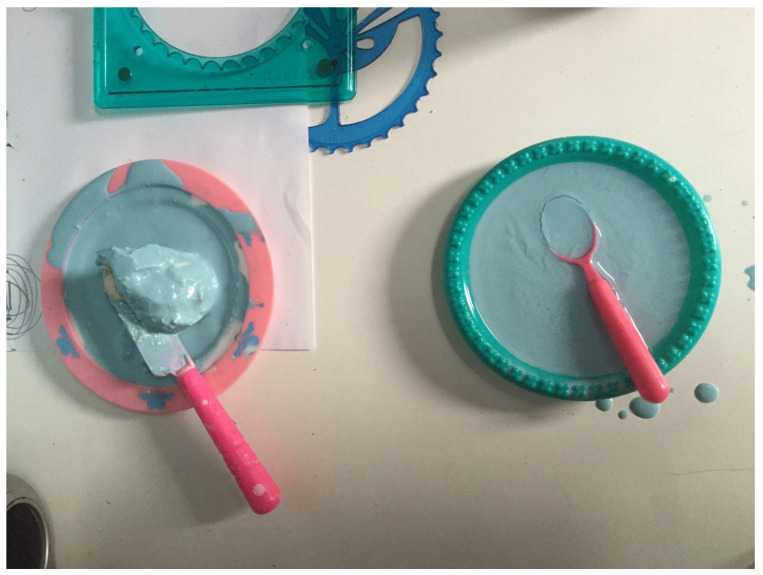
Art soup and meatballs (art as nourishment). Clay, plastic bowl and spoons, tempera paint.

**Figure 6 children-10-00751-f006:**
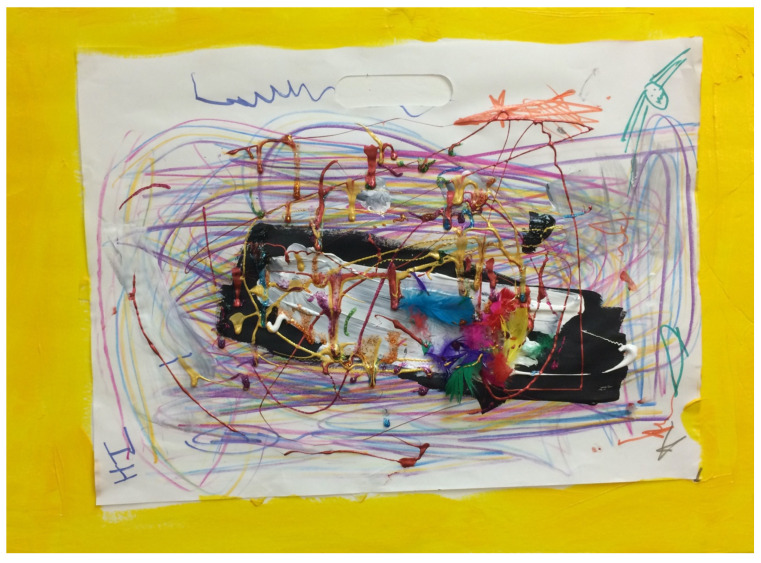
Bringing mom into the mess/Inside. 11 × 18″, colored paper, markers, acrylic paint, feathers.

**Figure 7 children-10-00751-f007:**
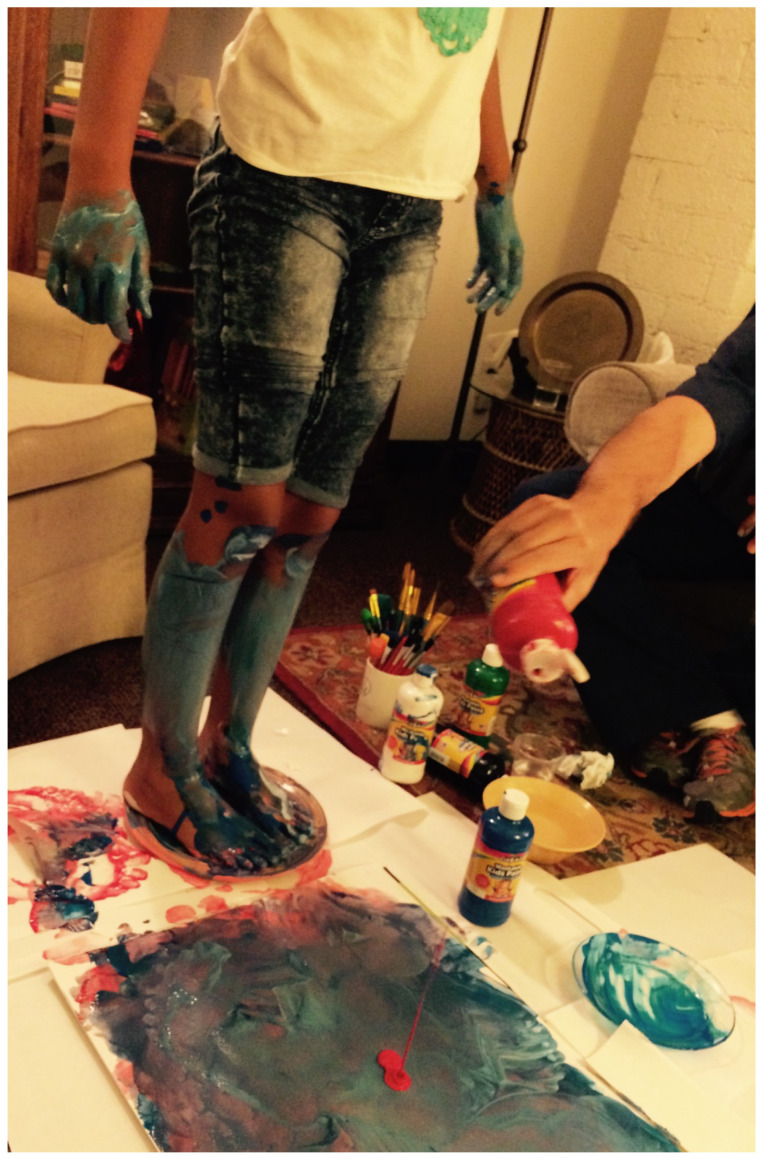
Will my dad support my needs? Large sheets of paper, tempera paint.

**Figure 8 children-10-00751-f008:**
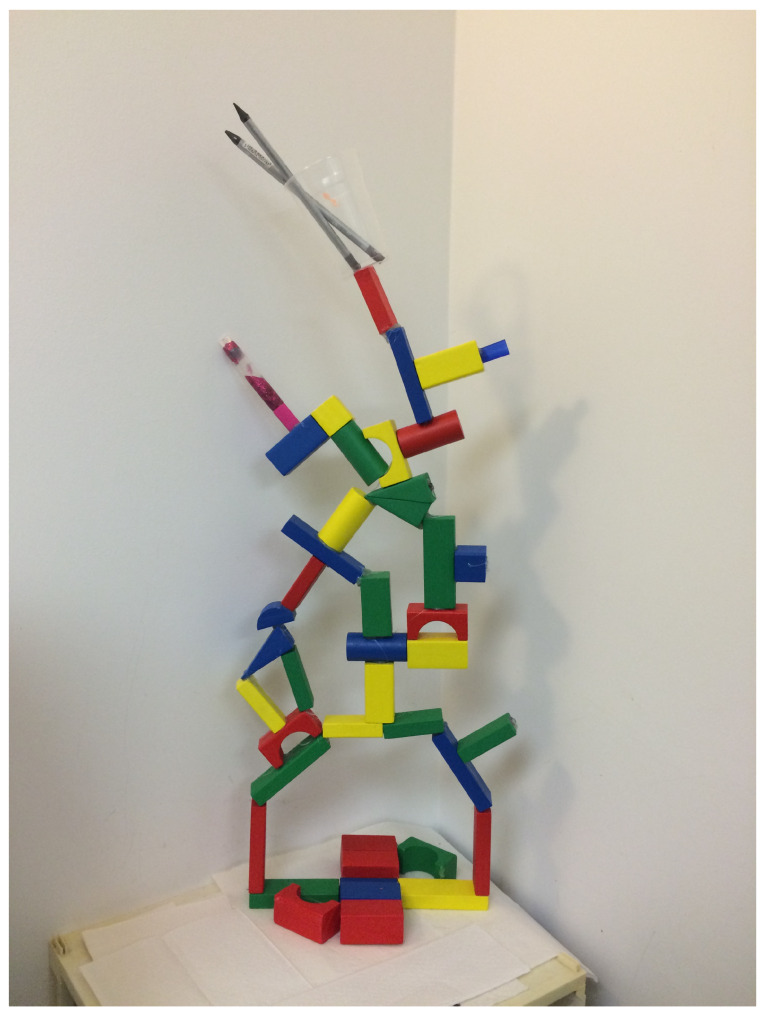
Building together something new (with dad). Wooden. blocks, glue gun, pencils, glitter tube.

**Figure 9 children-10-00751-f009:**
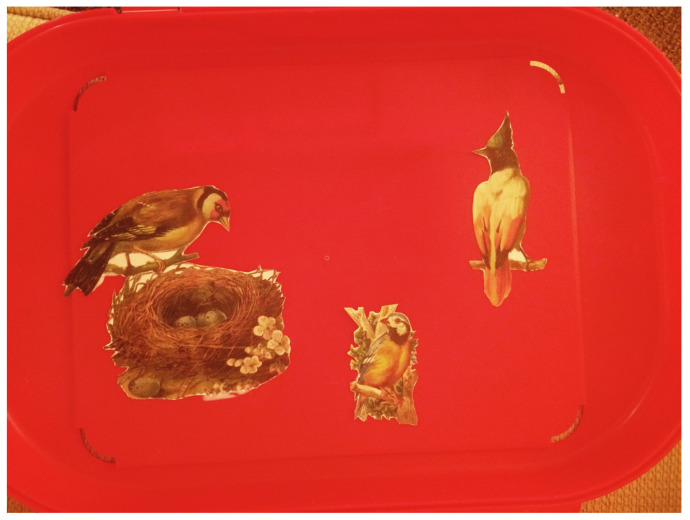
Birds and nests I carry with me/Transitional object. Plastic container, collage and stickers.

## Data Availability

Requests for supplementary data regarding this case can be requested directly from the author and in accordance with privacy and ethical restrictions of clinical work.

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
