# Peer review of "A Bird with No Name Was Born, Then Gone: A Child’s Processing of Early Adoption through Art Therapy"

_children, 2023, doi:10.3390/children10040751_

Round 1

Reviewer 1 Report

Thank you for the opportunity to read the article. However, the review was not easy for me. Although it is an interesting case study, it is not a professional article.

If it is to be a professional article, it would be good to:

1. improved the theoretical background of the problem (the influence of the substitute family care; the importance of art therapy - what it develops, principles, methods, ...),

2. supplement the methodological part - basic methodological data,

3. add conclusions and discussion - outputs from art therapy and their comparison with existing knowledge. 

Author Response

Responses to 

Reviewer # 1-

  1. improved the theoretical background of the problem (the influence of the substitute family care; the importance of art therapy - what it develops, principles, methods, ...) - I have attempted to revise by adding as requested about the influence of the substitute family (end of section 1.2) and about art therapy as a modality (beginning of section1.3), is not to explain what art therapy is or cover its overarching uses and features beyond illustrating its use in a particular case/ issue.
  2. supplement the methodological part - basic methodological data – I can attempt to clarify more how I reviewed the case. I’m unclear as to what “basic methodological data” the reviewer is missing in this case study…? I intentionally left out all unnecessary or identifiable information, of course, in order to protect the client’s privacy and confidentiality.
  3. add conclusions and discussion - outputs from art therapy and their comparison with existing knowledge. – I attempted to revise as requested.  

Thank you for your review.

Reviewer 2 Report

Dear author,

I appreciate your effort in preparing this case. I have some comments about some significant points;

- First, the manuscript should start by explaining the art therapy, its uses and features, and how it was adapted for adopted children. Then, the characteristic of the case and how the art therapy applied to this case as sessions should be given. This version was hard to follow on paper and needed to be scientifically sound. 

- Some paragraphs were too short and should be grouped with the other sections with some linking sentences (for instance, lines between 48-51). 

- Line 124-128 needs a reference.

- The author might bring together her/his experience under a subtitle because this case report sounded like a lived experience narrative. 

- The paper should be shorter. Also, there are a lot of pictures on paper, and I recommend selecting some of them. For example, Figures 1 and 2 are nearly the same; the author should give one of them. 

- The discussion section is too short; it is like a conclusion. The author should discuss the effect of art therapy in this case and for the adopted children with literature. 

- The references should be written according to journal rules. 

I wish you success in your work. 

Author Response

Responses to 

Reviewer #2 –

 - First, the manuscript should start by explaining the art therapy, its uses and features, and how it was adapted for adopted children. Then, the characteristic of the case and how the art therapy applied to this case as sessions should be given. This version was hard to follow on paper and needed to be scientifically sound. 

– I have attempted to further clarify and attend to the above. However, my intention is not to explain what art therapy is or cover its overarching uses and features. Rather, I intentionally began with the particular clinical challenge – namely, early adoption (sections 1.1 and 1.2) - and later addressed what art therapy with children who had been adopted entails (section 1.3) and the specific features of art therapy (section 1.4 - metaphors and transitional objects) which are often understood as central to the effectiveness of art therapy with children who had been adopted. Lastly, I’m not sure what was perceived as not “scientifically sound” in the literature review, or whether the expectation the reader has for “scientifically sound” applies to the attempt of systematically explore a particular case to illustrate the use of a modality (art therapy) and the psychological dynamics it allows clients to work through.

- Some paragraphs were too short and should be grouped with the other sections with some linking sentences (for instance, lines between 48-51). 

Corrected as requested.

- Line 124-128 needs a reference.

Corrected as suggested.

- The author might bring together her/his experience under a subtitle because this case report sounded like a lived experience narrative.

 – This is a point of contention for me; the whole intent of a case report as I see it is to offer a lived experience, in this case through narrative and art, related to the problem (and treatment in this case) at hand.  I attempted to tighten the narrative, added a bit about case study methodology and enhanced the discussion as requested with the hope that the usefulness of the narrative as a systematic review that develops clinical consideration for art therapy in other cases – is clearer.

- The paper should be shorter. Also, there are a lot of pictures on paper, and I recommend selecting some of them. For example, Figures 1 and 2 are nearly the same; the author should give one of them.

– I attempted to reduce the paper size a bit. However, there is an essential difference between figures 1 and 2, which align directly with the narrative: figure 1 zooms in and shows the detail of what becomes the core metaphor of the child, Figure 2 exhibits the box / next within its setting, which visually supports the reader in understanding how it was hanging, likely how it fell, and later vanished. This case study is based on art therapy works and narrates the child's experience through her art. I, therefore, ask to keep all pictures as part of the narrative.

- The discussion section is too short; it is like a conclusion. The author should discuss the effect of art therapy in this case and for the adopted children with literature. Added as requested.

- The references should be written according to journal rules. 

- - I had worked with an editor who followed the journal’s instructions for editing the references. Where are you seeing a problem…? I checked again and cannot see the issue. Happy to fix, just need clarification.

Thank you for your review!

Reviewer 3 Report

The article is very interesting but unfortunately requires a major revision.
The last keyword is not clear
The abstract needs to be better developed in a structured way: backgroud, objectives, method means, result conclusions
The introduction part is too long although very interesting. It is better to summarise all the background sections and if possible unify them.
The methodological part is absolutely insufficient. The tools used, in which context, for how long and the theoretical references to the specific methodologies must be clearly indicated. Furthermore, in the methodology the reason why first-person exposure is used should be explained. The definition of the case-study style used is missing.
The results are very interesting, but also very confusing because they are interspersed with both the description of the methodological part and the discussion part, i.e. they contain descriptions of the techniques used and comments on the products obtained. To make the entire system clear, it is necessary to separate the methodological description of the discussion and the results in the strict sense. Furthermore, it is necessary to better structure and summarise each individual result.
Given the confusing manner in which the results were presented, which engulfed much of the discussion, the section on the discussion of the results is very poor and insufficient.
The Conclusion section is absolutely missing.
The part describing the limitations of the research and future developments is absolutely missing.

Author Response

Reviewer #3 -

The article is very interesting but unfortunately requires a major revision.
The last keyword is not clear – I changed the last word to say “expressive” per suggestion
The abstract needs to be better developed in a structured way: background, objectives, method means, result conclusions attempted to revise and clarify as suggested
The introduction part is too long although very interesting. It is better to summarise all the background sections and if possible unify them. – I attempted to revise as requested
The methodological part is absolutely insufficient. The tools used, in which context, for how long and the theoretical references to the specific methodologies must be clearly indicated. Attempted to respond while also responding to other reviewers differing suggestions Furthermore, in the methodology the reason why first-person exposure is used should be explained.- explained The definition of the case-study style used is missing. - added
The results are very interesting, but also very confusing because they are interspersed with both the description of the methodological part and the discussion part, i.e. they contain descriptions of the techniques used and comments on the products obtained. To make the entire system clear, it is necessary to separate the methodological description of the discussion and the results in the strict sense. – I attempted to separate as much as possible, while keeping the relevance and validity of the narrative

Furthermore, it is necessary to better structure and summarise each individual result. – the results are all parts (themes) within a linked process (therapy journey), thus they cannot be separated or they will lose their meaning. I have nevertheless attempted to more clearly summarize before and after the case illustration the meaning of the results (themes) as they apply to art therapy considerations with adopted children.
Given the confusing manner in which the results were presented, which engulfed much of the discussion, the section on the discussion of the results is very poor and insufficient.  – based on this comment, and another reviewer’s suggestion, I augmented the discussion

The Conclusion section is absolutely missing. – I added a conclusion as requested

The part describing the limitations of the research and future developments is absolutely missing. – added (per this reviewer’s request; not typical for case studies as far as I know).

Thank you for you review!

Round 2

Reviewer 1 Report

The Changes helped to improve the article.

In this form, the Article may be published.

Congratulations to the author.

Reviewer 2 Report

The author has revised the paper systematically. The scientific soundness of the manuscript is better now. I appreciate your effort to rewrite the manuscript and explain the difference between the figures. The author also merged her experience and scientific knowledge in the discussion part. It is clearer now. 

I wish you success in your work. 

Reviewer 3 Report

The article can be published